# Measurement of the dynamic charge response of materials using low-energy, momentum-resolved electron energy-loss spectroscopy (M-EELS)

Sean Vig[1], Anshul Kogar[1], Matteo Mitrano[1], Ali A. Husain[1], Luc Venema[1], Melinda S. Rak[1], Vivek Mishra[2], Peter D. Johnson[3], Genda D. Gu[3], Eduardo Fradkin[1], Michael R. Norman[4] and Peter Abbamonte[1*]

**1** Department of Physics and Frederick Seitz Materials Research Laboratory, University of Illinois, Urbana, IL 61801, USA
**2** Oak Ridge National Laboratory, Oak Ridge, TN, 37831, USA
**3** Condensed Matter Physics and Materials Science Department, Brookhaven National Laboratory, Upton, NY, 11973, USA
**4** Materials Science Division, Argonne National Laboratory, Argonne, IL, 60439, USA

★ abbamonte@mrl.illinois.edu

## Abstract

One of the most fundamental properties of an interacting electron system is its frequency- and wave-vector-dependent density response function, $\chi(\mathbf{q}, \omega)$. The imaginary part, $\chi''(\mathbf{q}, \omega)$, defines the fundamental bosonic charge excitations of the system, exhibiting peaks wherever collective modes are present. $\chi$ quantifies the electronic compressibility of a material, its response to external fields, its ability to screen charge, and its tendency to form charge density waves. Unfortunately, there has never been a fully momentum-resolved means to measure $\chi(\mathbf{q}, \omega)$ at the meV energy scale relevant to modern electronic materials. Here, we demonstrate a way to measure $\chi$ with quantitative momentum resolution by applying alignment techniques from x-ray and neutron scattering to surface high-resolution electron energy-loss spectroscopy (HR-EELS). This approach, which we refer to here as "M-EELS", allows direct measurement of $\chi''(\mathbf{q}, \omega)$ with meV resolution while controlling the momentum with an accuracy better than a percent of a typical Brillouin zone. We apply this technique to finite-q excitations in the optimally-doped high temperature superconductor, $Bi_2Sr_2CaCu_2O_{8+x}$ (Bi2212), which exhibits several phonons potentially relevant to dispersion anomalies observed in ARPES and STM experiments. Our study defines a path to studying the long-sought collective charge modes in quantum materials at the meV scale and with full momentum control.

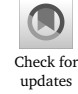
# 1 Introduction

An interacting electron system can often be described, at low energy scales, in terms of a set of weakly interacting, emergent particles [1, 2]. Such particles are usually either fermions, referred to as quasiparticles, or bosons, referred to as collective modes, though fractional or nonabelian particles may also emerge. The field of "quantum materials" might be defined as studies of these excitations at energy scales less than a few times room temperature, say, below 100 meV.

Outstanding experimental probes exist for studying both the quasiparticles and the spin collective modes. The former may be studied using angle-resolved photoemission (ARPES), which measures the one-electron spectral function, $A(\mathbf{k}, \omega)$, with meV-energy resolution and momentum accuracy of less than a percent of a Brillouin zone [3]. Quasiparticles may also be studied using scanning tunneling microscopy (STM), which measures a real-space spectral function that can be related to ARPES via Fourier transform coupled with models of quasiparticle scattering [4]. Spin collective modes may be studied with inelastic neutron scattering, traditionally using a triple-axis spectrometer [5], whose energy and momentum resolutions are similar to ARPES.

Surprisingly, there has never been an equivalent momentum-resolved probe of the charge collective modes in materials. The three commonly used finite-wavevector probes are neutrons, electrons and x-rays. As explained in Section 2, none of these techniques—as currently practiced—probes valence band charge excitations with both meV resolution and quantitative control over the momentum, $\mathbf{q}$.

Here, we demonstrate a strategy for measuring meV charge collective modes using momentum-resolved, low-energy electron energy-loss spectroscopy (M-EELS). Our strategy is to apply angular alignment techniques from x-ray and neutron scattering [5,6] to reflection high-resolution EELS (HR-EELS) [7,8], which is a meV-resolved probe of the collective charge excitations of a surface. We will show that it is possible, in this manner, to measure the dynamic charge response of a material, $\chi''(\mathbf{q}, \omega)$, with energy resolution close to 1 meV while controlling the momentum to an accuracy better than a percent of a typical Brillouin zone. As a case study, we apply M-EELS to the optimally doped high-temperature suprconductor,

$Bi_2Sr_2CaCu_2O_{8+x}$ (Bi2212), in which we observe collective modes relevant to the dispersion anomalies observed in ARPES [9,10] and STM [11] experiments, among other features. We argue that M-EELS will play a central role in spectroscopic studies of quantum materials in the coming decades.

This article is organized as follows. In Section 2, we explain the limitations of current momentum-resolved scattering techniques and why M-EELS, at the moment, is the best approach to studying the charge excitations at the meV scale. Section 3 describes our experimental approach, which combines alignment techniques from x-ray and neutron scattering with surface HR-EELS using cylindrical analyzers. In Section 4, we generalize the multiple scattering theory of Mills and co-workers [12,13] and show that the M-EELS cross section is proportional to the dynamic charge response, $\chi''(\mathbf{q}, \omega)$. Section 5 validates this cross section by comparing M-EELS studies of Bi2212 at $\mathbf{q} = 0$ to results from infrared spectroscopy. Section 6 demonstrates the momentum capabilities of M-EELS with elastic and inelastic maps of the Brillouin zone, in which static features such as the well-known structural supermodulation are visible [3]. Section 7 demonstrates a way to reconstruct the full dynamic susceptibility, $\chi(\mathbf{q}, \omega)$, from M-EELS data. Section 8 uses these results to analyze the dispersion anomalies (or "kinks") observed in ARPES experiments [9,10]. Section 9 summarizes the future prospects for M-EELS and the role it may play in spectroscopic studies of quantum materials in the future.

## 2 Why M-EELS?

Emergent particles in a many-electron system, in their simplest form, are described by three basic quantities [1,2]. The first, characterizing the fermions, is the one-electron Green's function,

$$G(\mathbf{r}, \mathbf{r}', t - t') = -i \left\langle \{ \psi^\dagger(\mathbf{r}, t), \psi(\mathbf{r}', t') \} \right\rangle \theta(t - t')/\hbar,$$

which represents the probability that an electron placed at spacetime location $(\mathbf{r}, t)$ will propagate to $(\mathbf{r}', t')$. $G$ quantifies the quasiparticle band structure, lifetimes, transport coefficients, etc. The second, characterizing bosons with charge character, is the dynamic density response

$$\chi_{\rho\rho}(\mathbf{r}, \mathbf{r}', t - t') = -i \left\langle [\hat{\rho}(\mathbf{r}, t), \hat{\rho}(\mathbf{r}', t')] \right\rangle \theta(t - t')/\hbar,$$

which represents the probability that a disturbance in the charge density at $(\mathbf{r}, t)$ propagates to $(\mathbf{r}', t')$. $\chi_{\rho\rho}$ characterizes the charge collective modes, such as plasmons. The third quantity is the dynamic spin response,

$$\chi_{SS}(\mathbf{r}, \mathbf{r}', t - t') = -i \left\langle [\hat{S}(\mathbf{r}, t), \hat{S}(\mathbf{r}', t')] \right\rangle \theta(t - t')/\hbar,$$

which characterizes spin collective modes, such as magnons.

We currently have outstanding, meV-resolved probes of both $G$ and $\chi_{SS}$. Angle-resolved photoemission spectroscopy (ARPES) measures the one-electron spectral function, $A(\mathbf{k}, \omega) = -Im[G(\mathbf{k}, \mathbf{k}, \omega)]/\pi$, where $G(\mathbf{k}, \mathbf{k}', \omega)$ is the Fourier transform of $G(\mathbf{r}, \mathbf{r}', t - t')$, probing the fermion quasiparticles with extraordinary energy and momentum resolution [3]. The fermions can also be measured in real space using scanning tunneling microscopy (STM), which measures the real-space spectral function $A(\mathbf{r}, \omega) = -Im[G(\mathbf{r}, \mathbf{r}, \omega)]/\pi$ [4].[1] Inelastic neutron scattering measures the dynamic spin response function, $\chi''_{SS}(\mathbf{q}, \omega) = Im[\chi_{SS}(\mathbf{q}, \mathbf{q}, \omega)]$, where $\chi_{SS}(\mathbf{q}, \mathbf{q}', \omega)$ is the Fourier transform of $\chi_{SS}(\mathbf{r}, \mathbf{r}', t - t')$ [14], probing the spin collective modes with similar resolution.

---

[1]Note that ARPES and STM are not Fourier transforms of one another, but may be related using models of quasiparticle scattering [4].

Unfortunately, there is no analogous probe of $\chi_{\rho\rho}$, at least at the meV scale. It is important to pause here and review the reasons why. What is needed is a momentum-resolved scattering technique that measures the dynamic charge response function, $\chi''_{\rho\rho}(\mathbf{q}, \omega) = Im[\chi_{\rho\rho}(\mathbf{q}, \mathbf{q}, \omega)]$, where $\chi_{\rho\rho}(\mathbf{q}, \mathbf{q}', \omega)$ is the Fourier transform of $\chi_{\rho\rho}(\mathbf{r}, \mathbf{r}', t - t')$. The three options for such probes are neutrons, electrons, and x-rays.

In the case of inelastic neutron scattering, the probe particle is electrically neutral and does not couple to charge excitations. Because of the nuclear cross section, neutrons can, of course, be used to study lattice excitations (phonons), which involve explicit displacements of the nuclear positions [5]. But electronic excitations, such as plasmons, cannot be studied with neutron techniques.

A more promising approach is inelastic electron scattering or "electron energy-loss spectroscopy" (EELS), which directly couples to charge exitations. The EELS cross section is, in the limit of zero temperature, given by the dielectric loss function, $-Im[1/\epsilon(\mathbf{q}, \omega)]$, which is proportional to $\chi''_{\rho\rho}(\mathbf{q}, \omega)$ [2]. EELS experiments may be carried out either in transmission or reflection geometry. The former requires high-energy ($\sim 10^5$ eV) electrons and has been done using both dedicated instruments [15, 16] and energy filters integrated with a scanning transmission electron microscope (STEM) [17, 18]. The latter is usually done using low-energy ($\sim 100$ eV) electrons and is normally used for surface science applications [7, 8].

The problem with EELS is that meV energy resolution has not yet been demonstrated in an instrument that also provides quantitative control over the momentum transfer, $\mathbf{q}$. Dedicated transmission EELS setups have excellent momentum resolution [15, 16], but have achieved at best 80 meV energy resolution [19] which, because of interference from the zero-loss line, makes them unsuitable for studying excitations in the sub-100 meV range. STEM instruments have achieved 18 meV resolution using $\Omega$-filters, but these setups are currently momentum-integrating, and Lorentzian tails of the elastic line obscure excitations at low energy [17, 18]. Surface EELS instruments can achieve energy resolution of 1 meV or better [7, 8], but have not been implemented in a manner that allows the momentum transfer to be determined with high accuracy (see below and Section 3). Some variant on high-energy EELS employing aberration correctors seems likely to be the best long-term strategy for studying meV charge collective modes. But these techniques are still a work in progress.

The last option is inelastic x-ray scattering (IXS). Carried out at 3rd-generation synchrotron facilities, IXS techniques simultaneously achieve high momentum resolution and sub-meV energy resolution using backscattering Si analyzers [20]. While IXS should, in principle, be capable of studying valence charge excitations, it is not practical for doing so, for the following subtle reason. The x-ray cross section is proportional to $\chi''_{nn}(\mathbf{q}, \omega) = Im[\chi_{nn}(\mathbf{q}, \mathbf{q}, \omega)]$, where $\chi_{nn}(\mathbf{q}, \mathbf{q}', \omega)$ is the Fourier transform of the propagator for the electron density [21],

$$\chi_{nn}(\mathbf{r}, \mathbf{r}', t - t') = -i \left\langle [\hat{n}(\mathbf{r}, t), \hat{n}(\mathbf{r}', t')] \right\rangle / \hbar.$$

Unfortunately, the quantity of interest in a real material is not the electron density propagator, $\chi_{nn}$, but the charge density propagator, $\chi_{\rho\rho}$. These two quantities are not the same, because the positively charged nucleii in a solid contribute to the charge density, $\rho$, but not to the electron density, $n$. For example, the integrated charge density of an electrically neutral atom is zero, while its integrated electron density is equal to $Z$, the number of electrons, the vast majority of which reside in core states [22].

The types of excitations that contribute to $\chi_{\rho\rho}$ and $\chi_{nn}$ are therefore fundamentally different. $\chi_{\rho\rho}$ reveals excitations that modulate the charge density of the system, e.g. valence plasmons and, in the case of ionic materials, phonons. Neutral excitations, such as phonons in covalent solids like Si or Ge, modulate the charge density very little and contribute to $\chi_{\rho\rho}$ only to the extent that they modulate the valence electron density. $\chi_{nn}$, on the other hand, exhibits excitations that modulate the electron density. Because most of the electrons in a solid reside

in core states, $\chi_{nn}$ is overwhelmingly dominated by phonons, which displace the atomic cores. Valence excitations that leave the atomic positions fixed, such as plasmons, contribute to $\chi_{nn}$ but are weaker by a factor of $1/Z$.

In practice, what this means is that meV-resolved IXS, while sensitive to charge excitations in principle, in practice is essentially a phonon technique. The sum rules on the IXS response function $\chi_{nn}''(\mathbf{q}, \omega)$, in cases of interest, are mostly exhausted by the lattice excitations. Electronic excitations in the spectra are swamped by the lattice modes, which are stronger by a factor of $Z$. For this reason, meV-resolved IXS has been extraordinarily successful at mapping phonon dispersion relations [20, 23], even in crystals that are too small to be studied with neutron techniques. But it is an inefficient way to study the valence charge excitations of fundamental interest in a many-electron system.

For this reason, x-ray researchers have turned to resonance techniques. By tuning the x-ray beam energy to a core absorption edge, scattering from valence excitations can be greatly enhanced, an approach referred to as resonant inelastic x-ray scattering or "RIXS" [24, 25]. Using this approach, researchers have been able to detect valence excitations not visible with nonresonant IXS [26,27]. Despite steady improvements, however, the best resolution achieved with RIXS is still only about $\Delta E \sim 40$ meV [28], and fundamental questions remain about whether the RIXS cross section can be related to a well-defined response function [25]. In the long run, RIXS is sure to make a major impact on our understanding of the collective excitations in materials, but at the moment it is not practical for studying the collective modes at the sub-100 meV scale.

In summary, we currently have no truly momentum-resolved way of measuring one of the most fundamental properties of a many-body system, $\chi_{\rho\rho}(\mathbf{q}, \omega)$. In addition to characterizing charge collective modes, $\chi_{\rho\rho}(\mathbf{q}, \omega)$ is the charge susceptibility of the system, which quantifies its response to external fields [1, 2], as well as its tendency to exhibit charge order [29]. In the limit $\mathbf{q} \to 0$ and $\omega \to 0$, $\chi_{\rho\rho}(\mathbf{q}, \omega)$ also quantifies the electronic compressibility of the system [1, 2]. The imaginary part, $\chi_{\rho\rho}''(\mathbf{q}, \omega)$, is also related to the inverse dielectric or "loss" function of the system, $-Im[1/\epsilon(\mathbf{q}, \omega)]$, providing information about the screening properties at $\mathbf{q} \neq 0$. The absence of an experimental probe of the density response means that these utterly basic phenomena are unknown for the vast majority of materials.

Here, we demonstrate that the dynamic charge response of a material, and hence the valence charge excitations, can be measured with both meV resolution and quantitative momentum control using by applying alignment techniques from x-ray and neutron scattering to reflection high-resolution EELS (HR-EELS). We refer to this approach as "M-EELS". In this technique, a monochromatic beam of low-energy electrons (10 eV $< E <$ 200 eV) is scattered from the surface of a material in ultrahigh vacuum [7]. The scattered electrons are detected using an electrostatic energy analyzer. Using aberration-corrected cylindrical optics, an energy resolution better than 0.5 meV has been achieved [30]. Historically, HR-EELS has been thought of as a surface science technique, often for studying vibrations of molecular adsorbates [31], and has not been applied widely in the field of quantum materials. But the information it provides should directly complement that from ARPES and STM, which are also surface techniques.

Using HR-EELS for momentum-resolved studies of collective modes presents two challenges. The first is multiple-scattering. In the case of transmission EELS, the single-scattering approximation is often valid and the Born cross section is directly proportional to $\chi_{\rho\rho}''(\mathbf{q}, \omega)$. In the case of reflection EELS, strong interaction with the sample surface causes the electrons to scatter many times before reaching the detector. Fortunately, as shown in the early 1970's by Mills and co-workers [12, 13], this multiple scattering problem is soluble: The amplitude for elastic scattering of low-energy electrons is much larger than that for inelastic scattering, so the surface EELS problem may be solved using the distorted wave Born approximation (DWBA) [32]. In Section 4, we will show that Mills' solution implies that reflection EELS

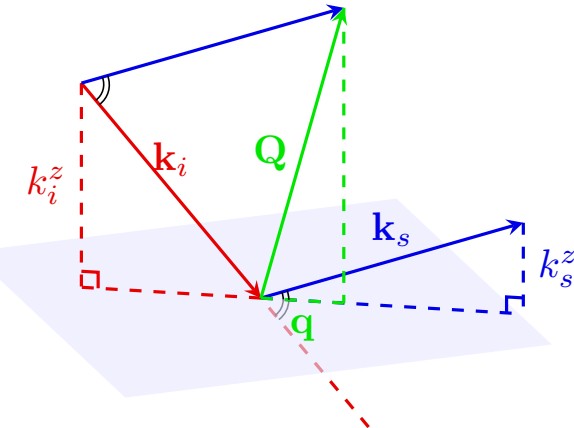

Figure 1: Schematic showing the M-EELS scattering geometry. Here, $\mathbf{k}_i$ and $\mathbf{k}_s$ are the momenta of the incident and scattered electron, respectively, and $\mathbf{Q}$ is the momentum transfer, the in-plane component of which, $\mathbf{q} = (q_x, q_y)$, is the quantity of interest in M-EELS. The out-of-plane momentum components, $k_i^z$ and $k_s^z$, enter the scattering matrix elements in a critical way (see Section 4 and Appendix A).

measures the density-density correlation function of a surface, $S(\mathbf{q}, \omega)$, which is directly proportional to $\chi_{\rho\rho}''(\mathbf{q}, \omega)$ via the fluctuation-dissipation theorem.

The second complication is that HR-EELS experiments have not been done in a manner that allows quantitative control over the momentum transfer, $\mathbf{q}$ (Fig. 1). In true, wave-vector resolved techniques, such as neutron and x-ray scattering [5], the sample is mounted on a diffractometer whose multiple axes of rotation are aligned to intersect at a single point to a precision of a few tens of microns. The alignment errors are characterized by a volume called the "sphere of confusion," which sets the overall momentum accuracy of the instrument. This volume is centered on the probe beam using a set of translations that is separate from those used to align the rotation axes. Using a third set of translations mounted on top of the goniometer stack, the sample is placed in this same location, and oriented by measuring the elastic scattering from at least two, noncolinear Bragg reflections of the crystal. The angles of these reflections are used to construct an orientation matrix relating the diffractometer motions to momentum space, enabling momenta to be indexed with a precision of thousandths of a reciprocal lattice unit [6]. Without such alignment, it is still possible to observe dispersion effects simply by rotating the sample [33]. But phenomena requiring accurate momentum definition, such as studies of the critical fluctuations near a phase transition or the Goldstone modes of an ordered phase [34, 35], require a more sophisticated approach.

Here, we describe a strategy for combining meV-resolution HR-EELS techniques with single-crystal alignment techniques widely used in x-ray and neutron scattering. This hybrid approach allows direct measurement of the dynamic charge response function, $\chi_{\rho\rho}''(\mathbf{q}, \omega)$, and hence studies of the long-sought meV charge collective modes in solids, with meV resolution and momentum accuracy better than one percent of a typical Brillouin zone. In the ensuing discussion we will drop the subscripts and refer to the charge response function simply by its conventional name, $\chi''(\mathbf{q}, \omega)$.

## 3 Experimental approach

There are three practical ways one might approach implementing an M-EELS experiment. The first is to use a single-point, aberration-corrected Ibach spectrometer [7], widely used

in surface science, mating it to a multi-axis sample goniometer and control system with the degrees of freedom necessary to mimic a triple-axis spectrometer used for inelastic neutron or x-ray scattering [5]. The advantages of this approach are that high resolution, $\Delta E \sim 0.5$ meV, has already been demonstrated [30], and that alignment and data collection protocols from inelastic neutron scattering can be implemented in a straight-forward way. The disadvantage of this approach is that data collection is slow, as the spectrometer samples only one $(\mathbf{q}, \omega)$ point at a time.

The second approach is to use a variant on an Ibach spectrometer that provides parallel energy detection, i.e., a cylindrical analyzer with a position-sensitive detector [36]. This approach is faster, in principle, since a full energy spectrum may be collected in parallel. But aberration-free focusing combined with high throughput is more challenging to achieve in the analyzer in this configuration.

The third approach is to combine an Ibach-type electron gun [7] with an ARPES hemispherical analyzer [37, 38]. This approach provides, in principle, the fastest data collection rate, since it samples a complete wedge of momentum and energy space in parallel. The disadvantage is experimental complexity: In triple axis spectroscopy, the momentum transfer, $\mathbf{q}$, is quantified by precisely measuring the angle between the scattered electrons and the direct beam. Reflection EELS measurements must, however, be carried out at a scattering angle greater than $\sim 60^o$, so that appreciable in-plane momenta can be reached without horizon problems from the sample surface. The angular acceptance of a hemispherical analyzer is typically only about $30^o$, so the incident and scattered electrons cannot both be measured in a fixed experimental geometry. The electron gun itself must, therefore, be placed on a rotation stage [37], so the angle between the beam and the analyzer can be adjusted, introducing many alignment complications as well as significantly elevated cost compared to other approaches.

While the hemispherical approach has been implemented before [37, 38], we focused on the use of cylindrical analyzers, evaluating both single-point and parallel detection schemes. On the basis of stability, reproducibility, cleanliness of the elastic line (i.e., compactness of the tails of the resolution function), and throughput, we found the best data were provided by the single-point approach, despite its slow data collection speed. It is this approach that we describe here.

Our setup is based on a commercially available, aberration-corrected, surface HR-EELS spectrometer with a double-pass monochromator and single-pass analyzer, whose ultimate resolution is $\sim 1$ meV. The analyzer angle rotation (called "two-theta") was modified to reduce mechanical backlash and actuated with a stepper motor. The spectrometer resides inside a magnetically shielded, ultrahigh vacuum (UHV) chamber pumped with a cryopump and a $LN_2$-cooled titanium sublimation pump (TSP), exhibiting a base pressure of $5 \times 10^{-11}$ torr and a residual field at the sample position of $\sim 3$ mG. The spectrometer is typically run at 4 meV resolution, which provides a direct-beam current of 140 pA at the detector.

This system is mated to a custom, low-temperature sample goniometer consisting of a differentially-pumped rotary seal, which acts as the primary sample rotation (called "theta"), and two independent sets of XY translations—one below and one above the seal. The former provides the motions needed to align the axis of rotation of theta to that of two-theta, and the latter allows placement of the sample in this position. An out-of-plane ("phi") rotation is achieved using a $360^o$ piezo rotator. Two cameras oriented at $90^o$ vantage points monitor the sample through holes in the magnetic lining, facilitating alignment (see below). Cooling is achieved using a standard He flow cryostat connected to the sample via a set of $e$-beam-welded Cu braids, providing a base temperature of 17.5 K without radiation shields. With the braids in place, the range of motion of phi is limited to $100^o$. Surface preparation is carried out in a separate chamber equipped with a LEED system and annealing stage, though most surfaces are prepared simply by cleaving.

True momentum space scanning is enabled by a custom control system based on SPEC, a crystal orientation package commonly used on synchrotron beamlines. The vendor-supplied control system was replaced by a programmable microcontroller that mediates communications between the host and the voltage box of the EELS, enabling energy and momentum scanning via coordinated control of the goniometer angles, scattering angle, and lens voltages.

Proper alignment of the rotations and sample orientation is crucial for momentum-resolved experiments. The beam generated by the electron gun was aligned by the manufacturer to within 100 $\mu$m of the axis of rotation of two-theta, which is adequate for momentum studies of samples $\sim 0.3$ mm or larger. The challenge is to align the rotation axis of theta, as well as the sample itself, to this point. This is done by the following procedure. First, the zero of two-theta is set by scanning the detector through the direct beam. Next, using the XY sample motions, a reference sample is translated into the center of rotation of theta by viewing it from the video cameras (a flat, cleaved graphite crystal works well for alignment purposes). This sample is then lowered into the spectrometer and the specular beam reflected into the analyzer. The theta angle of this reflection is then optimized for a series of two-theta values over the range $[0^o, 70^o]$. If the axes are perfectly centered, a plot of theta vs. two-theta will form a line with slope 1/2. If the axes are misaligned, this plot will exhibit some curvature, quantified by a quadratic term in a polynomial fit. Using the second set of XY translations, the rotary seal is translated parallel to the beam until this quadratic component is negligible, i.e., the curve is linear to within the angular resolution of the instrument, which is $\sim 0.1^o$, at which point the axes may be considered centered. Once this procedure is complete, the material of interest can be aligned simply by placing it in the center of rotation of theta using the cameras.

After centering, the orientation of the crystal axes with respect to the goniometer angles still must be determined. The specular reflection is often not suitable for this purpose, since the cleavage surface may not be perfectly flat. Running the analyzer in zero-loss mode to select the elastic scattering, the crystal is rotated to identify two, noncollinear Bragg reflections of the surface, typically (1,0) and (0,1) (note that momenta can be indexed using two Miller indices, $(H, K)$, since momentum in the direction perpendicular to the surface is not conserved). These two reflections are then used to define the orientation matrix [6], at which point the system is ready for experiments. In this article, momenta will be denoted as illustrated in Fig. 1.

Once the above alignment steps have been completed, the momentum performance of an M-EELS experiment is quantified by two figures of merit: the momentum accuracy and resolution. The former describes the reliability with which $\mathbf{q}$ can be positioned in the Brillouin zone, and is determined by the size of the sphere of the confusion. The latter describes the size of the ellipsoid over which the instrument integrates at any given $\mathbf{q}$, and is given by the angular resolution of the spectrometer. The experiments in this article were carried out at 50 eV beam energy with a sphere of confusion of $\sim 0.25$ mm and a sample-slit distance of 70 mm, which translates to a momentum accuracy of $\sim (0.25/70)\sqrt{2mE}/\hbar = 0.013 \text{Å}^{-1}$. The angular resolution of our HR-EELS spectrometer is $\sim 17$ mrad, which implies a momentum resolution of $0.017\sqrt{2mE}/\hbar = 0.06 \text{Å}^{-1}$. A detailed description of the momentum resolution of a M-EELS experiment analogous to that developed for neutron spectrometers [39, 40] will be the subject of a future article.

# 4   M-EELS cross section and the dynamic susceptibility, $\chi(\mathbf{q}, \omega)$

As discussed in Section 2, a dominant effect in low-energy, reflection EELS is multiple scattering, which prevents the electrons from penetrating the material and causes them to couple only to excitations near the surface. A key insight for the technique, due to Mills and co-workers,

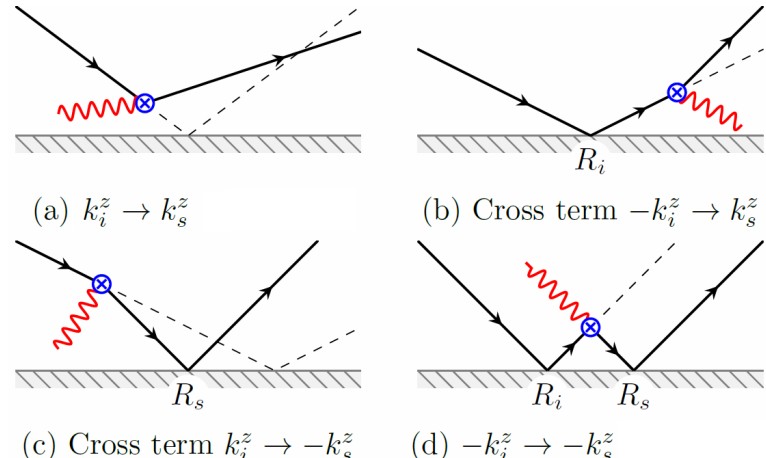

(a) $k_i^z \to k_s^z$

(b) Cross term $-k_i^z \to k_s^z$

(c) Cross term $k_i^z \to -k_s^z$

(d) $-k_i^z \to -k_s^z$

Figure 2: (a-d) Four different quantum mechanical processes captured in Mills' theory of surface EELS using the distorted wave Born approximation. The cross section is dominated by processes (b) and (c) (see Appendix A).

was that multiple scattering takes place almost entirely in the elastic channel [8,12,13]. That is, of the many scattering events an electron undergoes before reaching the detector, typically only one is inelastic. In such situations, the multiple scattering problem can be solved to a high degree of accuracy using the distorted wave Born approximation (DWBA) [32]. In this approach, the incident and final state plane wave functions of the probe electron are replaced with phenomenological wave functions that model reflectivity from the sample surface. In terms of these effective wave functions, the inelastic event can then be treated in the first Born approximation. Four possible scattering processes result, illustrated in Fig. 2. As originally argued by Mills, the cross section is dominated by the two terms that involve a single reflectivity event [12,13].

The only elastic scattering effect considered in Mills' original treatment was specular reflectance off the sample surface [12,13]. A crystal is periodic, however, so elastic Bragg scattering can also take place. In momentum-resolved studies, one often encounters such reflections, which reside at the center of each Brillouin zone. Here, we extend the Mills approach to the case of a periodic surface, considering how Bragg scattering modifies multiple scattering effects in the cross section.

The generalized M-EELS cross section in the presence of elastic scattering from Bragg planes with reciprocal vectors, $\mathbf{G}$, is derived in detail in Appendix A. The main result is

$$\frac{\partial^2 \sigma}{\partial \Omega \partial E} = \sigma_0 \sum_{\mathbf{G}} V_{\text{eff}}^2(\mathbf{q}-\mathbf{G}) \int_{-\infty}^{0} dz_1 dz_2$$

$$\times e^{-|\mathbf{q}-\mathbf{G}||z_1+z_2|} \left[ \sum_{m,n} \langle m|\hat{\rho}^*(\mathbf{q}-\mathbf{G}, z_1)|n\rangle \langle n|\hat{\rho}(\mathbf{q}-\mathbf{G}, z_2)|m\rangle P_m \delta(E - E_n + E_m) \right]. \quad (1)$$

Here, $\hat{\rho}(\mathbf{q}-\mathbf{G})$ is the density operator, $P_m = e^{-E_m/k_B T}/Z$ is the Boltzmann factor, and $V_{\text{eff}}(\mathbf{q}-\mathbf{G})$, is an effective Coulomb propagator that describes the interaction between the probe electron and the excitations near the surface of the semi-infinite system.

This result simplifies by recognizing that the quantity in the brackets is the two-point, density-density correlation function, i.e., the dynamic structure factor [2,41],

$$S(\mathbf{q}, z_1, z_2, \omega) = \sum_{m,n} \left[ \langle m|\hat{\rho}(\mathbf{q}, z_1)|n\rangle \cdot \langle n|\hat{\rho}(-\mathbf{q}, z_2)|m\rangle P_m \right] \delta(E - E_n + E_m). \quad (2)$$

This quantity is also sometimes called the Van Hove function [42]. This mixed representation form of $S$ is appropriate for a system in which momentum is conserved only in a two-dimensional plane, and characterizes density fluctuations with in-plane wave vector, $\mathbf{q}$, correlated between depths $z_1$ and $z_2$ below the surface. In terms of this quantity, the cross section becomes

$$\frac{\partial^2 \sigma}{\partial \Omega \partial E} = \sigma_0 \sum_{\mathbf{G}} V_{\text{eff}}^2(\mathbf{q}-\mathbf{G}) \cdot \int_{-\infty}^{0} dz_1 dz_2 e^{-|\mathbf{q}-\mathbf{G}||z_1+z_2|} S(\mathbf{q}, z_1, z_2, \omega), \tag{3}$$

where we have used the fact that the correlation function should exhibit the same periodicity as the material itself, i.e., $S(\mathbf{q}-\mathbf{G}, z_1, z_2, \omega) = S(\mathbf{q}, z_1, z_2, \omega)$.

Eq. 3 has three important implications. The first is that the probe depth of M-EELS is determined by the magnitude of the in-plane momentum transfer, $\mathbf{q}$. At momenta in the vicinity of the $\Gamma$ point ($|\mathbf{q}| \sim 0$), the scattering is dominated by the $\mathbf{G} = 0$ term in the sum, and the effective probe depth is $\sim 1/|\mathbf{q}|$, which is typically a few tens of nm or less. Hence, we see that M-EELS is a surface probe, but is somewhat more bulk sensitive than probes like ARPES or STM, which measure only the top layer of the material. This can be understood by realizing that, while the probe electron does not itself penetrate into the material, it feels the density fluctuations below the surface, which generate a long-ranged Coulomb potential extending into the vacuum. This conclusion, which follows directly from the original Mills treatment [12, 13], is likely the reason substrate phonons were visible in a recent study of FeSe thin films on $SrTiO_3$ [43]. In any case, we see that the core observable of M-EELS is the density-density correlation function of the surface, defined as

$$S_S(\mathbf{q}, \omega) = \int_{-\infty}^{0} dz_1 dz_2 e^{-|\mathbf{q}-\mathbf{G}||z_1+z_2|} S(\mathbf{q}, z_1, z_2, \omega) \approx S(\mathbf{q}, 0, 0, \omega). \tag{4}$$

The second implication of Eq. 3 is that M-EELS measures the long-sought density response function, which is related to the correlation function via the quantum mechanical version of the fluctuation-dissipation theorem [1, 2, 41, 44],

$$S_S(\mathbf{q}, \omega) = -\frac{1}{\pi} \frac{1}{1 - e^{-\hbar\omega/k_B T}} \chi_S''(\mathbf{q}, \omega), \tag{5}$$

where $\chi_S''(\mathbf{q}, \omega)$ is the density response function of the surface, $(1 - e^{-\hbar\omega/k_B T})^{-1} = n(\omega) + 1$ is a Bose factor that reflects the quantum statistics of the excitations. $\chi_S''(\mathbf{q}, \omega)$ is the imaginary part of the surface density propagator, $\chi_S(\mathbf{q}, \omega)$, which describes the propagation of collective charge excitations in the plane of the surface. This quantity is not precisely the same as the bulk $\chi(\mathbf{q}, \omega)$, but is the relevant quantity for comparison to ARPES and STM experiments, which also probe the surface of a material. Moreover, as we show in Section 5, $\chi_S''(\mathbf{q}, \omega)$ is very similar to the bulk response, $\chi''(\mathbf{q}, \omega)$, in cases where the latter can be measured by other techniques such as infrared spectroscopy and transmission EELS.

Third, Eq. 3 indicates that the Coulomb matrix element, $V_{\text{eff}}^2(\mathbf{q}-\mathbf{G})$, is energy-independent and diverges whenever the electron beam satisfies a Bragg condition (see Appendix A). In one respect, this makes interpretation of M-EELS data simpler than ARPES, whose matrix elements depend strongly on energy [3]. However, the M-EELS matrix elements are strongly momentum-dependent, the divergences having the effect of amplifying the intensity of inelastic scattering when the momentum coincides with a structural periodicity in the material. We refer to this phenomenon as "Bragg enhancement." A specific case is the well-known enhancement of the intensity when $\mathbf{q} \sim 0$, which in HR-EELS was traditionally referred to as the regime of "dipole scattering" [7, 8]. The M-EELS matrix elements have the effect of enforcing the periodicity of reciprocal space onto the experimental data, creating a finite-momentum replica of the electronic response near $\mathbf{q} = 0$ around each reciprocal lattice vector, $\mathbf{G}$.

The divergences of $V_{eff}^2(\mathbf{q} - \mathbf{G})$ imply that there must be corrections to the Mills DWBA approach in near-Bragg conditions, since the experimental intensity must remain finite. Such corrections are beyond the scope of this paper. We point out, however, that it is highly likely that the matrix elements would continue to be energy-independent, even when Mills' theory breaks down. This opens up the possibility of correcting for multiple scattering effects by using frequency sum rules [45].

In the low-momentum limit, i.e., $\mathbf{q} \sim 0$, only the $\mathbf{G} = 0$ term in Eq. 3 is significant. In this limit, all other terms in Eq. 3 may be dropped, and the cross section reduces to [12, 13, 44],

$$\frac{\partial^2 \sigma}{\partial \Omega \partial E} = \sigma_0 V_{eff}^2(\mathbf{q}) \int_{-\infty}^{0} dz_1 dz_2 e^{-|\mathbf{q}||z_1 + z_2|} \cdot S(\mathbf{q}, z_1, z_2, \omega), \tag{6}$$

where

$$V_{eff}(\mathbf{q}) = \frac{4\pi e^2}{q^2 + (k_i^z + k_s^z)^2}. \tag{7}$$

This is exactly Mills' result, cited before in many previous works [7, 8, 12, 13, 44].

# 5 Comparison to IR and transmission EELS measurements

We now demonstrate the M-EELS approach by applying it to an optimally-doped high temperature superconductor, $Bi_2Sr_2CaCu_2O_{8+x}$ (Bi2212) with $T_c = 92$ K. Elastic momentum maps were carried out at an incident beam energy of 50 eV with resolution $\Delta E = 5$ meV, and high-resolution measurements of excitations were done at an energy of 7.4 eV with resolution $\Delta E = 2.2$ meV. Bi2212 was chosen for initial M-EELS studies because of its excellent cleavability, which facilitates surface preparation, and because it exhibits a structural supermodulation whose reflections are useful for defining the crystal orientation. For simplicity, in this article we will label momentum space in terms of the reduced, tetragonal unit cell, i.e., the Miller indices, $(H, K)$, denote a transferred momentum $\mathbf{q} = 2\pi(H, K)/a$, $a = 3.81$Å being the tetragonal, in-plane lattice parameter. We located and optimized the elastic scattering from both the (1,0) fundamental Bragg peak as well as the (0.11, 0.11) supermodulation reflection. The goniometer angles of these two reflections were used to construct an orientation matrix, allowing precise definition of the momentum transfer, $\mathbf{q}$ [6].

We begin by validating the cross section discussed in Section 4 and Appendix A. The dynamic charge response measured with M-EELS, $\chi''(\mathbf{q}, \omega)$, should be proportional to the dielectric loss function, $-Im[1/\epsilon(\mathbf{q}, \omega)]$ [48]. So the cross section expression in Eq. 3 can be evaluated by comparing M-EELS spectra at $\mathbf{q} \sim 0$ to results from infrared reflectivity (IR) measurements.

In Fig. 3a we show $\mathbf{q} = 0$ M-EELS measurements of Bi2212 for $\omega < 100$ meV at two different temperatures. For comparison, this plot also shows the inverse dielectric function determined from $c$-axis polarized, IR spectroscopy [46]. The most pronounced feature in the M-EELS spectra is a series of phonons that were previously observed in early, conventional HR-EELS studies [49–53]. Note that the energies of these modes, which reside at 17 meV, 24 meV, 49 meV, and 80 meV, closely coincide with the kink-like fermion dispersion anomalies observed in Bi2212 with ARPES, suggesting they may be related to these effects [9, 10]. Apart from a background visible in the M-EELS spectrum, which arises from in-plane excitations not seen in $c$-axis optics experiments, the two techniques are quantitatively consistent, both in terms of the energies and the relative oscillator strengths of the modes. This comparison validates Eq. 3 and shows that the surface response function, $\chi_S''(\mathbf{q}, \omega)$ is quite representative of that of the bulk, $\chi''(\mathbf{q}, \omega)$.

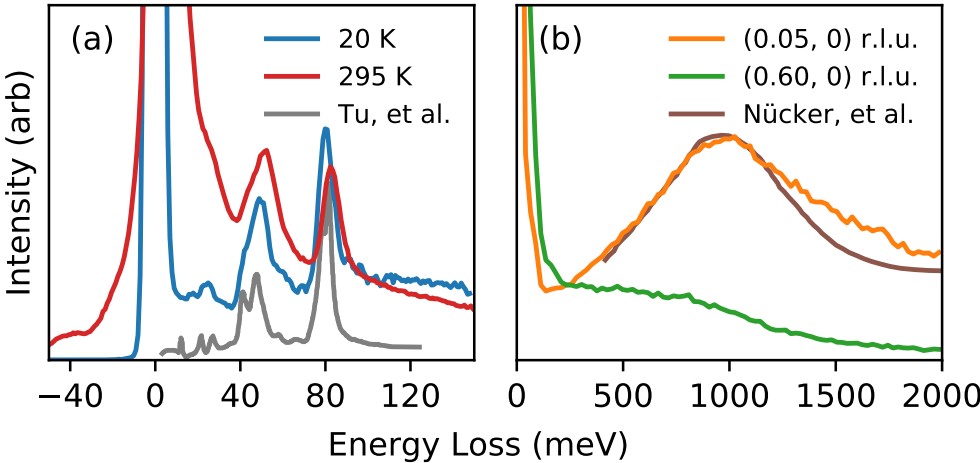

Figure 3: (a) Comparison of the M-EELS spectrum of Bi2212 at $\mathbf{q} = 0$ at $T = 20$ K (blue) and $T = 295$ K (red) to the $c$-axis loss function measured at $T = 295$ K with infrared spectroscopy, reproduced from Ref. [46] (gray). The energies and spectral weights of the optical phonons are quantitatively consistent between the two techniques. (b) Comparison of the M-EELS spectrum at $\mathbf{q} = (0.05, 0)$ (orange) to transmission EELS data at $q = 0.08$ r.l.u. reproduced from Ref. [47] (brown). The energy and linewidth of the 1 eV plasmon are, again, quantitatively consistent between the two techniques. At a large momentum, $\mathbf{q} = (0.6, 0)$, the plasmon damps into a broad, electronic continuum extending to a cutoff of 0.9 eV (green).

The purpose of M-EELS is to observe electronic excitations, such as plasmons. In Fig. 3b, we show the M-EELS spectrum in the plasmon region, i.e., the energy range $0 < \omega < 2$ eV, taken at 50 eV beam energy at a momentum transfer $\mathbf{q} = (0.05, 0)$. This spectrum is compared with that from a low-resolution, transmission EELS measurement performed at the same momentum value [47]. A clear plasmon excitation is visible at approximately 1 eV energy loss whose lineshape is very similar in the two spectra. At large momentum, $\mathbf{q} = (0.6, 0)$, this plasmon decays into a particle-hole continuum that bears a strong resemblance to the electronic Raman continuum observed in inelastic light scattering experiments on cuprates [54, 55]. These measurements demonstrate that M-EELS is sensitive to the electronic charge excitations that are so difficult to observe with x-ray or neutron techniques.

One of the striking conclusions to draw from Fig. 3 is that M-EELS appears to be quantitatively consistent with bulk measurements, at least at $\mathbf{q} \sim 0$. Both IR spectroscopy and transmission EELS probe bulk excitations, with a probe depth of $\sim 100$ nm, the former determined by the extinction depth of IR light and the latter by the thickness of the suspended films used for experiments. Evidently, as discussed in Section 4, the M-EELS probe depth is actually reasonably large, despite the fact the electrons themselves do not penetrate the material.

## 6 Momentum maps

The crystal orientation capabilities of M-EELS enable measurements to be carried out with quantitative control over the momentum transfer, $\mathbf{q}$. In Fig. 4, we show a static, fixed-energy map of momentum space of Bi2212 taken at room temperature. This measurement was performed with an energy resolution $\Delta\omega = 4.5$ meV at zero energy-loss ($\omega = 0$) and a fixed out-of-plane momentum, $L = 20.3$, by doing coordinated scans of the sample and analyzer angles. The matrix elements, $V_{eff}^2$, were divided from the raw data, making the assumption

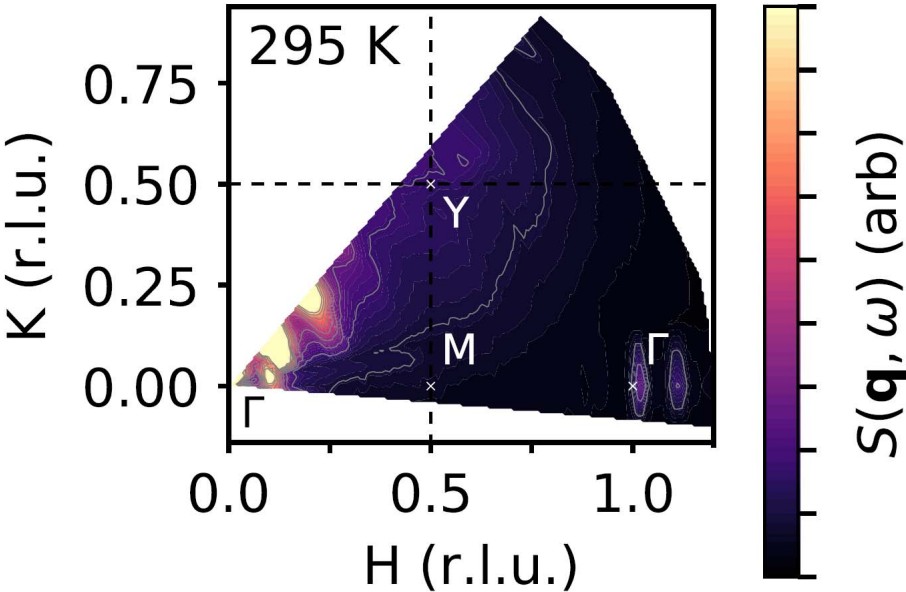

Figure 4: Momentum maps of the quasi-static ($\omega = 0$) scattering in Bi2212 in the $(H, K)$ plane at $T = 295$ K.

that $\mathbf{G} = 0$ is the dominant term (Eq. 6). The data in Fig. 4 are therefore nominally proportional to the correlation function, $S(\mathbf{q}, \omega)$. The fundamental, static density features are visible, including the (1,0) Bragg peak and the structural supermodulation reflection (as well as its harmonics) that were used to define the sample orientation. This map demonstrates the ability of M-EELS to quantitatively pinpoint charge features in momentum space.

A key advantage of M-EELS is its energy resolution. In Fig. 5b we show energy- and momentum-dependent M-EELS spectra at $T = 20$ K for four different sections of momentum space, denoted $(i)$-$(iv)$ as defined in Fig. 5a. These data were taken with energy resolution $\Delta E = 2.2$ meV at a fixed out-of-plane momentum $L = 1.9$ r.l.u., and show the raw intensity without dividing out the matrix elements. These sections show the dispersion of the phonons (Fig. 3a) using a custom color scale so both elastic and inelastic features can be seen on the same plot. The data in Fig. 5 were taken on a different sample than Fig. 4, so also serve as a reproducibility check.

A prominent feature in these spectra is a set of bright, vertical lines (labeled SM) that indicate enhanced inelastic intensity at wave vectors corresponding to the structural supermodulation. This effect, which we call "Bragg enhancement," is a consequence of the Coulomb matrix elements in Eq. 3, which enhance the scattering when $\mathbf{q}$ coincides with a structural periodicity in the material. Because the matrix elements in M-EELS are energy-independent, this effect could in principle be normalized out using a sum rule. Fig. 5 illustrates that the phonons (Fig. 3a) are weakly dispersive, in quantitative agreement with HR-EELS studies [53]. This is additional evidence that these excitations may be related to the ARPES dispersion kinks, whose energy shape is consistent with nondispersive, Einstein-like modes [56]. Note that, unlike other materials in which acoustic phonons are clearly visible [57], no acoustic modes are visible in Bi2212 using M-EELS.

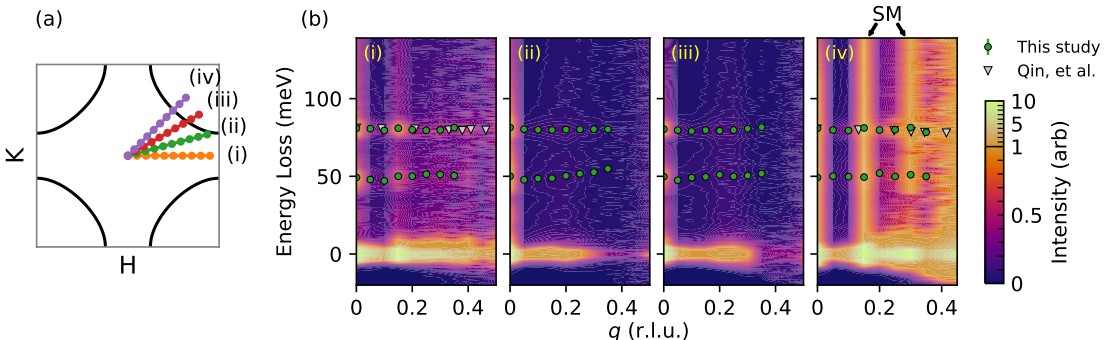

Figure 5: (a) Brillouin zone of Bi2212 indicating the four momentum space trajectories used for M-EELS studies, labeled (i)-(iv). (b) Momentum-dependent M-EELS spectra of optimally-doped Bi2212 at 20 K for the four momentum directions indicated in panel (a). A custom color scale was devised so that strong features around the zero-loss line can be observed on the same figure as weak phonon features (see scale bar). The bright, vertical lines labeled "SM" are a consequence of the Coulomb matrix element, $V_{\text{eff}}^2(\mathbf{q}-\mathbf{G})$, which is enhanced when $\mathbf{q}$ coincides with a structural periodicity in the material, in this case the structural supermodulation. The dispersions of the optical phonons (green points) are consistent with previous studies (empty triangles) reproduced from Ref. [53].

# 7    Reconstructing $\chi(\mathbf{q}, \omega)$

The most important application of M-EELS, in the long run, may be in determining the dynamic susceptibility, $\chi(\mathbf{q}, \omega)$. As discussed in Section 2, this quantity is the fundamental charge propagator of the system, characterizing its electronic compressibility, its response to external fields, its tendency to exhibit charge order, and its ability to screen charge [1, 2, 29]. Here, we sketch out a procedure for using M-EELS to quantify the full $\chi(\mathbf{q}, \omega)$ for a real material. We will base our analysis on the data in Fig. 5b. Our goal is not to construct a susceptibility function that is exact, but to illustrate a practical procedure for an approximate function that is based on real M-EELS data. In Section 8, we will show how this quantity can be used to analyze self-energy effects in ARPES data.

The starting point for this procedure is Eqs. 3-5. The former relates the experimental intensity to the correlation function, $S_S(\mathbf{q}, \omega)$, and the latter relates $S_S$ to the imaginary part of the response, $\chi_S''(\mathbf{q}, \omega)$, which we have shown bears great similarity to $\chi''(\mathbf{q}, \omega)$ of the bulk.

Following Eq. 3, the first step is to divide the matrix elements, $V_{\text{eff}}^2$, from the experimental data, to obtain the correlation function. Doing so raises two caveats. The first is that the quantity achieved in this manner does not have the proper units, i.e., is only proportional to $S_S(\mathbf{q}, \omega)$. No information about the absolute value of $S_S$ is available from M-EELS data. The absolute scale could, in principle, be calibrated by applying a frequency sum rule [45], an approach commonly used in IXS [58]. But a sum rule for the surface correlation function measured with M-EELS has not yet been derived.

The second caveat is that the expression for the cross section (Eq. 3) is only approximate and breaks down in the limit $q \to 0$ and $\omega \to 0$ due to multiple scattering effects not included in Mills' theory (see Section 4). This is evident from the functional form of $V_{\text{eff}}^2$, which diverges in this limit, even though the experimental intensity must remain finite. The consequence is that the $S_S(\mathbf{q}, \omega)$ obtained by dividing out $V_{\text{eff}}^2$ will vanish at small energy and momentum with a functional form that differs from the expected asymptotic properties of a correlation function. Multiple scattering corrections to Mills' theory would have to be implemented for a simple division of the matrix element to be meaningful in this limit.

Nevertheless, the matrix elements for M-EELS are energy-independent, and it is likely they would remain so even in an exact scattering theory. What this means is that the M-EELS spectrum should exhibit the same frequency dependence as the correlation function, $S_S(\mathbf{q}, \omega)$, at all values of $\mathbf{q}$, even in the regime where $V_{\text{eff}}^2$ diverges. The only quantity that is unknown is the overall normalization. Hence, if a sum rule were derived for M-EELS, it might be used to correct for multiple scattering effects even in the low-momentum region. Efforts to acquire such a sum rule are in progress.

We divided the experimental spectra (Fig. 5b) by $V_{\text{eff}}^2$, to achieve a discrete representation of $S_S$ over a complete, reduced octant of the Brillouin zone. In doing so, we made the approximation that the sum is dominated by the $\mathbf{G} = 0$ term, i.e., that the elastic scattering from the surface is dominated by the specular reflection. This approximation should be valid everywhere except in very close proximity to the supermodulation reflections.

Having acquired an approximate form for $S_S(\mathbf{q}, \omega)$, the next step is to determine $\chi_S''(\mathbf{q}, \omega)$ from the fluctuation-dissipation theorem (Eq. 5). The main difference between $\chi_S''$ and $S_S$ is that the former is antisymmetric in $\omega$, while the ratio between positive and negative energy features in the latter is determined by the temperature. Acquiring $\chi_S''$ from $S_S$ therefore requires eliminating the Bose factor in Eq. 5, but this too comes with caveats: This factor is singular in the limit $\omega \to 0$, and depends explicitly on the sample temperature, which is not always known exactly. The most stable way to determine $\chi''$ from $S$ is to antisymmetrize, by making use of the identity,

$$\chi_S''(\mathbf{q}, \omega) = -\pi [S_S(\mathbf{q}, \omega) - S_S(\mathbf{q}, -\omega)]. \tag{8}$$

The advantage of this expression is that it seamlessly handles the singularity at $\omega = 0$ and does not require knowledge of the sample temperature. Application of Eq. 8 to the M-EELS spectrum at $\mathbf{q} = 0$ is illustrated in Fig. 6a. Note that the result is perfectly antisymmetric, by construction, but exhibits anomalies in the vicinity of $\omega = 0$ due to the experimental resolution, which leads to violation of Eq. 5 in the low-energy region [44]. These anomalies are intrinsic to M-EELS in the sense that their scale can be reduced by improving the experimental resolution, but they can never be eliminated completely. We antisymmetrized our data for all momenta at which our function $S_S(\mathbf{q}, \omega)$ is defined, resulting in an approximate representation of $\chi_S''$ over one octant of the Brillouin zone.

With $\chi_S''(\mathbf{q}, \omega)$ in hand, the real part of the susceptibility, $\chi_S'(\mathbf{q}, \omega)$, was determined using a Kramers-Kronig transform. This required extrapolating each energy spectrum, which was measured up to 150 meV energy loss, using a $1/\omega^2$ tail [60]. Finally, we assumed the Brillouin zone has four-fold symmetry and repeated $\chi$ to fill all momentum space. The full dynamic susceptibility function is shown in Fig. 7. The static, momentum-dependent susceptibility, $\chi_S(\mathbf{q}, 0)$, is shown in Fig. 6b for comparison. The strong momentum-dependence of this latter quantity may suggest a tendency of this material to form charge order [29, 61].

## 8   Self-energy effects

We close by illustrating the usefulness of $\chi_S''(\mathbf{q}, \omega)$ for analyzing fermion dispersion anomalies observed in ARPES experiments. The primary 49 and 80 meV phonon modes observed with M-EELS (Fig. 3a) have the correct energy and dispersion to explain the dispersion kinks observed with ARPES in Bi2212 [9, 10]. To make a quantitative comparison, we used our experimentally determined susceptibility function, $\chi_S(\mathbf{q}, \omega)$, to compute the lowest order correction to the electron self-energy in a one-loop approximation, which is given by the convolution inte-

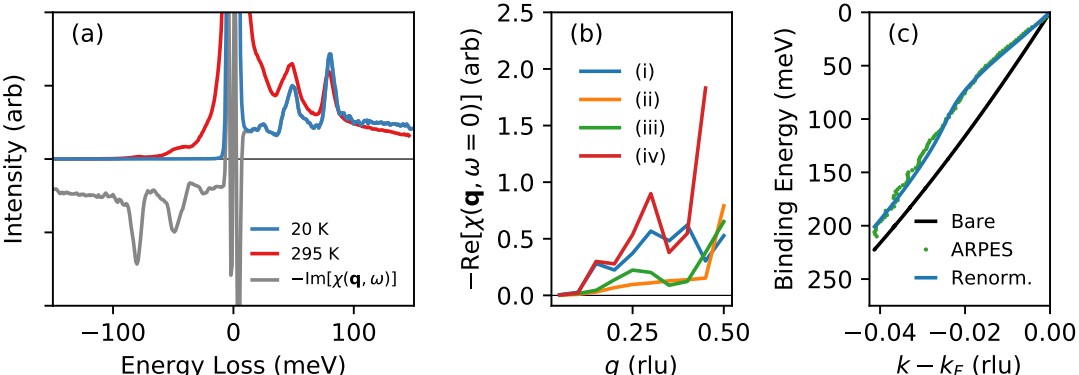

Figure 6: (a) Illustration of the antisymmetrization procedure for determining $\chi''(\mathbf{q}, \omega)$ from the correlation function, $S(\mathbf{q}, \omega)$. The energy asymmetry of the M-EELS spectrum at $\mathbf{q} = 0$ at 295 K (red) and 20 K (blue) is a consequence of the Bose factor (Eq. 5). The antisymmetrized spectrum representing $\chi''(\mathbf{q}, \omega)$ is shown in gray. (b) Momentum dependence of the static susceptibility, $\chi(\mathbf{q}, 0)$, at $T = 20$ K for the four momentum directions defined Fig. 5a. Note that $\chi(\mathbf{q}, 0)$ is purely real. (c) The calculated bare (black) and renormalized (blue) band dispersion of Bi2212 are compared to the measured ARPES data (Ref. [59]). The renormalized electron self-energy was determined using a one-loop calculation using the experimentally determined $\chi''(\mathbf{q}, \omega)$.

gral [1],

$$\Sigma(i\omega, k) = g_{bf}^2 \, T \sum_{\Omega} \int d\mathbf{q} \, G_0(i\omega - i\Omega, \mathbf{k} - \mathbf{q}) \chi_S(i\Omega, \mathbf{q}). \tag{9}$$

Here, $\chi$ is the charge propagator determined from M-EELS, $G_0$ is the bare electron Green's function, and $\omega$ and $\Omega$ are Matsubara frequencies for the fermions and bosons, respectively. The variables $\mathbf{k}$ and $\mathbf{q}$ are the associated momenta, $T$ is the temperature, and $g_{bf}$ is an effective boson-fermion coupling constant, which in this expression is assumed to be momentum-independent.

In order to avoid complications related to opening of the superconducting gap [9, 10], we focus here on the ARPES data in the nodal direction for $T = 115$ K $> T_c$. $G_0$ was taken from tight binding fits to ARPES data at high binding energies [62], which gave a bare Fermi velocity of $v_F = 1.7$ eVÅ used as a seed value in data fits.

It is necessary to allow for the possibility that different phonon modes exhibit different electron-phonon coupling constants. For this purpose, we made an analytic parameterization of $\chi_S$ by fitting the M-EELS results at all momenta with two Lorentzians, for the 49 and 80 meV modes. This allowed evaluation of Eq. 9 using a different value for $g_{bf}$ for each mode. The M-EELS data used to determine the susceptibility were measured at $T = 20K$. But the spectrum exhibited no observable temperature dependence, so it is reasonable to use it to analyze ARPES data even at $T = 115$ K. After evaluating Eq. 9, real frequency self-energy was then determined by analytic continuation.

Using the two electron boson coupling constants, the Fermi velocity, and the overall magnitude of $\chi_S''(\mathbf{q}, \omega)$ as parameters, we fit the low-energy ARPES dispersion along the nodal direction [59]. The best fit was obtained for coupling constants $g_{bf1} = g_{bf2} = 0.5$ eV for the 49 meV and 80 meV modes, respectively, resulting in the dispersion curves in Fig. 6c. The agreement between the calculated and experimental curves is excellent and gives an estimate for the energy- and momentum-integrated electron-phonon coupling constant of $\lambda = v_F / v_F^0 - 1 = 0.7$, which is in line with previous estimates [63]. We conclude that the bosonic modes observed

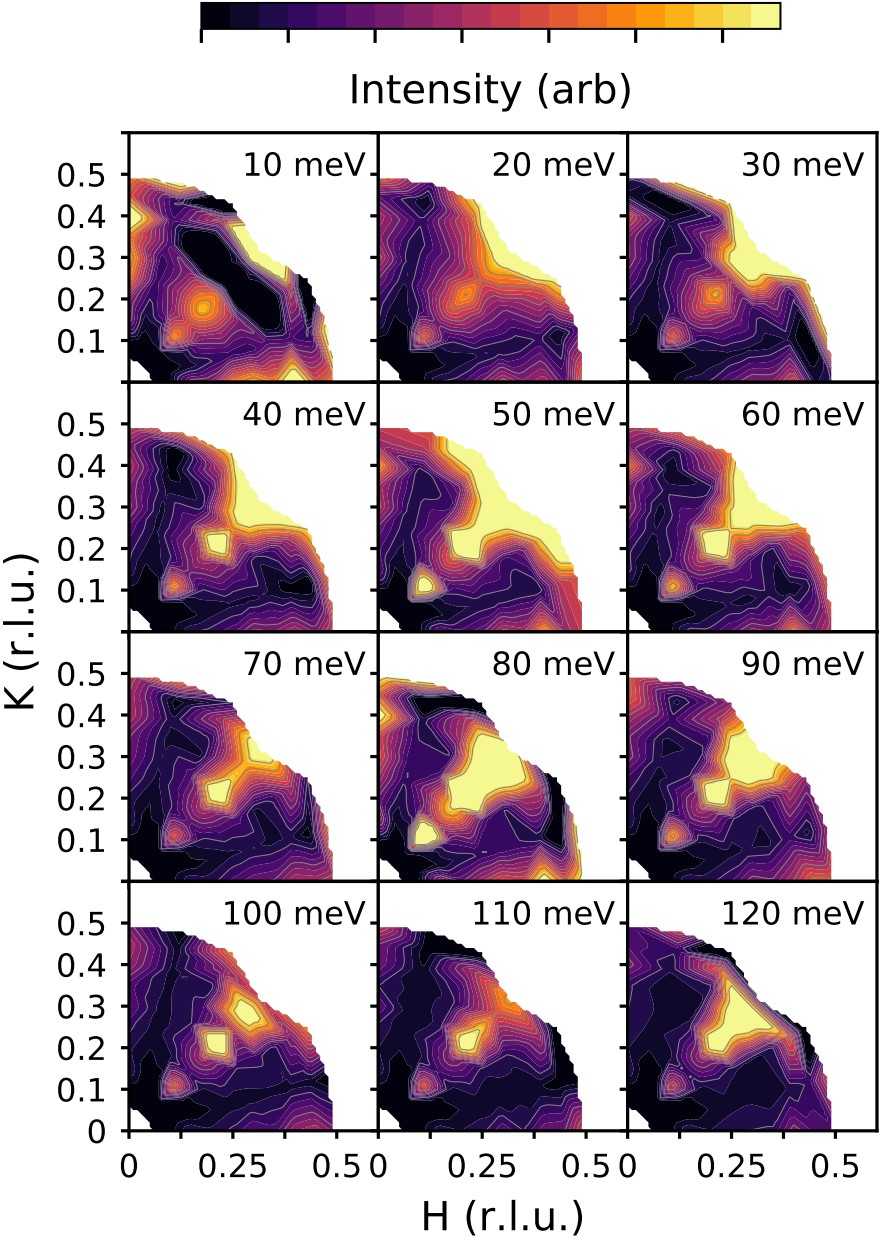

Figure 7: Fixed-energy momentum cuts of $\chi''(\mathbf{q}, \omega)$ at $T = 20$ K for different values of $\omega$. The data have been multiplied by $q^2$ to improve the visibility of high-momentum features. Bragg enhancement due to the scattering matrix elements is visible as enhanced weight in the region of the structural supermodulation.

with M-EELS have the correct energy structure to explain the ARPES kinks, and may be part of the cause of these effects.

This one-loop calculation is simplistic, and only provides an estimate of $\lambda$. It also does not exclude the possibility that the dispersion kinks could arise, at least in part, from spin fluctuations or other intrinsic, many-body effects. It should certainly not be taken as a claim that phonons are the mechanism of high temperature superconductivity in cuprates. But this exercise illustrates the point that M-EELS data can be tremendously useful for understanding boson effects observed in one-electron spectroscopies, including not only ARPES but also STM [64].

## 9 Future prospects for M-EELS

A great deal of work remains to be done, particularly on the subjects of sum rules and quantifying multiple scattering effects beyond the Mills framework. However, it should be clear from this study that M-EELS is a direct and, at the moment, unique way to measure the dynamic charge susceptibility and the bosonic charge modes at the meV scale in condensed matter. We expect this technique to take its place alongside ARPES, STM, and neutron scattering as one of the fundamental wave-vector-resolved probes of elementary excitations in quantum materials. In the long run, meV-resolved, high-energy EELS, carried out in transmission geometry with extraordinarily high energy resolution using aberration correctors, may someday extend the range of applicability of these techniques to achieve bulk information on this energy scale.

## Acknowledgements

We gratefully acknowledge helpful input from L. H. Santos, B. Uchoa, J. M. Tranquada, C. M. Varma, J. C. Davis, and J. Zaanen.

**Funding information**  This work was supported by the Center for Emergent Superconductivity, an Energy Frontier Research Center funded by the U.S. Department of Energy, Office of Basic Energy Sciences under Award DE-AC02-98CH10886. P. A. acknowledges support from the EPiQS program of the Gordon and Betty Moore Foundation, grant GBMF4542. E. F. acknowledges DOE Award No. DE-SC0012368. M. M. acknowledges support from the Alexander von Humboldt Foundation.

## A Derivation of the M-EELS cross-section

A detailed derivation of the scattering cross section for low-energy EELS measurements has been presented previously by several authors [8, 12, 13, 44]. However, these apply only to the case in which the surface is translationally invariant, and ignore the possibility of elastic Bragg scattering off the in-plane crystalline structure. Here, we generalize these results to the case of a periodic system. Like past treatments [8, 13, 44], we base our analysis on the distorted-wave Born approximation (DWBA) method [32].

The physical justification for DWBA in this case is that multiple scattering in reflection EELS predominantly takes place in the elastic channel, while inelastic events may be treated in the first Born approximation. Following the treatment of Mills and coworkers [8, 12, 13], the elastic scattering can be described by using the phenomenological wave functions,

$$\psi_i = N_i\left[e^{i\mathbf{k}_i\cdot\mathbf{r}}e^{ik_i^z z} + \sum_{\mathbf{G}_1}R_{\mathbf{G}_1}e^{i(\mathbf{k}_i+\mathbf{G}_1)\cdot\mathbf{r}}e^{-i\kappa_i z}\right]\theta(z), \tag{10}$$

$$\psi_s = N_s\left[e^{i\mathbf{k}_s\cdot\mathbf{r}}e^{ik_s^z z} + \sum_{\mathbf{G}_2}R_{\mathbf{G}_2}e^{i(\mathbf{k}_s+\mathbf{G}_2)\cdot\mathbf{r}}e^{-i\kappa_s z}\right]\theta(z), \tag{11}$$

where $\psi_i$ and $\psi_s$ represent the incident and scattered electron, respectively. In this ansatz, the coordinate $\mathbf{R} = (\mathbf{r}, z)$, where $\mathbf{r}$ is the in-plane component and $z$ is the component normal to the surface. $\mathbf{k}_{i,s}$ and $k_{i,s}^z$ represent, respectively, the in-plane and out-of-plane momenta of the incident and scattered electron. The parameter $R_{\mathbf{G}}$ is the complex amplitude reflection coefficient for the surface Bragg reflection with wave-vector $\mathbf{G}$, the specular reflection being denoted by $\mathbf{G} = 0$. $N_{i,s}$ are the appropriate normalization constants.

In Bragg scattering from a surface, the in-plane component of the electron momentum changes by $\mathbf{G}$. In order to conserve energy, the out-of-plane momentum must take on the value $\kappa = \sqrt{k_z^2 - G^2 - 2\mathbf{k}\cdot\mathbf{G}}$, where $k_z$ was its momentum before scattering. This momentum change can always be accomplished near a surface since momentum is not conserved in the $z$ direction. Hence, the reflected, out-of-plane momenta in Eqs. 1 and 10 are given by

$$\kappa_{i,s} = \pm\sqrt{(k_{i,s}^z)^2 - G^2 - 2\mathbf{k}_{i,s}\cdot\mathbf{G}}, \tag{12}$$

where the sign is chosen so that $\kappa_{i,s}$ has the same sign as $k_{i,s}^z$.

In EELS one usually neglects exchange effects, which is equivalent to assuming that spin-flip scattering from magnetic excitations is negligible (such magnetic scattering is referred to as SPEELS [65]). The scattering matrix element, then, is just given by the direct Coulomb term,

$$M_{n,m} = -\frac{ie^2}{2\hbar}\int d\mathbf{R}_1 d\mathbf{R}_2 \frac{\langle n|\hat{\rho}(\mathbf{R}_1)|m\rangle \psi_s^*(\mathbf{R}_2)\psi_i(\mathbf{R}_2)}{|\mathbf{R}_1 - \mathbf{R}_2|} \tag{13}$$

where $\mathbf{R}_1$ and $\mathbf{R}_2$ are the valence and probe electron coordinates, respectively. Neglecting exchange scattering effectively renders these two electrons distinguishable. $|m\rangle$ and $|n\rangle$ are the initial and final many-body states of the material, and $\hat{\rho}$ is the density operator.

Inserting Eqs. 10-11 into Eq. 13 yields four independent scattering processes, illustrated in Fig. 2. As pointed out by Mills [12,13], the dominant terms are those that involve a single reflectivity event, i.e., those shown in Fig. 2b and 2c. Keeping only these two terms, and integrating over coordinates $\mathbf{r}_1$, $\mathbf{r}_2$, and $z_2$, the matrix element becomes

$$M_{n,m} = -\frac{i}{2\hbar}N_s N_i \sum_{\mathbf{G}}\int_{-\infty}^{0} dz_1 V_{2D}(\mathbf{q}-\mathbf{G})\cdot$$
$$\left[\frac{R_G^*}{|\mathbf{q}-\mathbf{G}| - i(k_i^z + \kappa_s)} + \frac{R_{-G}}{|\mathbf{q}-\mathbf{G}| + i(\kappa_i + k_s^z)}\right]\langle n|\hat{\rho}(\mathbf{q}-\mathbf{G}, z_1)|m\rangle e^{-|\mathbf{q}-\mathbf{G}||z_1|} \tag{14}$$

where $V_{2D}(\mathbf{q}) = 2\pi e^2/q$ is the two-dimensional Coulomb propagator. From this matrix element, we can compute the transition rate using Fermi's golden rule,

$$\omega_{n\leftarrow m} = 2\pi\hbar|M_{n,m}|^2. \tag{15}$$

For the case of a perfectly coherent electron beam, squaring the matrix element (Eq. 14) results in many cross terms involving two different $\mathbf{G}$ values. For an incoherent source, such as the thermionic source in a typical M-EELS setup, these cross terms will average to zero [41]. If we drop these cross terms and assume the system exhibits inversion symmetry, i.e., $R_{\mathbf{G}} = R_{-\mathbf{G}}$, the transition rate simplifies to

$$\omega_{n\leftarrow m} = 2\pi\frac{1}{\hbar}(N_s N_i)^2 \sum_{\mathbf{G}}|R_{\mathbf{G}}|^2 V_{\text{eff}}^2(\mathbf{q}-\mathbf{G})\cdot$$
$$\int_{-\infty}^{0} dz_1 dz_2 e^{-|\mathbf{q}-\mathbf{G}||z_1+z_2|}\langle m|\hat{\rho}^*(\mathbf{q}-\mathbf{G}, z_1)|n\rangle\langle n|\hat{\rho}(\mathbf{q}-\mathbf{G}, z_2)|m\rangle \tag{16}$$

where $V_{\text{eff}}^2(\mathbf{q}-\mathbf{G})$ is a Coulomb matrix element given by

$$V_{\text{eff}}^2(\mathbf{q}-\mathbf{G}) = \frac{V_{2D}^2(\mathbf{q}-\mathbf{G})}{2}\left[\frac{4|\mathbf{q}-\mathbf{G}|^2 + (k_i^z + k_s^z + \kappa_s + \kappa_i)^2 + (k_i^z + \kappa_s - \kappa_i - k_s^z)^2}{\left(|\mathbf{q}-\mathbf{G}|^2 + (\kappa_i + k_s^z)(k_i^z + \kappa_s)\right)^2 + |\mathbf{q}-\mathbf{G}|^2(k_i^z + \kappa_s - \kappa_i - k_s^z)^2}\right]. \tag{17}$$

Note that if the beam satisfies a Bragg condition, then $\mathbf{q} = \mathbf{G}$, $k_i^z + \kappa_s - \kappa_i - k_s^z = 0$ and $k_i^z + \kappa_s + \kappa_i + k_s^z = 2(k_i^z + k_s^z)$, which implies that the denominator in Eq. 17 vanishes. We refer to this amplification of the M-EELS cross section under conditions of strong elastic scattering as "Bragg enhancement" (see Section 4).

From the transition rate, one can determine the double-differential scattering cross section using the standard relation

$$\frac{\partial^2 \sigma}{\partial \Omega \partial E} = \frac{1}{\Phi} \sum_{n,m} \omega_{n \leftarrow m} P_m \frac{\partial^2 N}{\partial \Omega \partial E} \tag{18}$$

where $\Phi = \sqrt{2E_i/m}/V$ is the electron flux, $P_m = \exp(-E_m/k_B T)/Z$ is the Boltzmann factor, and the last term is the density of final states,

$$\frac{\partial^2 N}{\partial \Omega \partial E} = \frac{V}{8\pi^3} \left(\frac{2m}{\hbar^2}\right)^{3/2} \sqrt{E_f}. \tag{19}$$

where $E_f = E_i - \hbar\omega$ is the energy of the scattered electron. Multiplying out these terms, putting in the energy-conserving delta functions, we arrive at the full expression for the generalized M-EELS cross section for the case of a periodic surface,

$$\frac{\partial^2 \sigma}{\partial \Omega \partial E} = \sigma_0 \sum_{\mathbf{G}} V_{\text{eff}}^2(\mathbf{q} - \mathbf{G}) \int_{-\infty}^0 dz_1 dz_2$$
$$\times \left[ \sum_{m,n} \langle m|\hat{\rho}^*(\mathbf{q}-\mathbf{G},z_1)|n\rangle \langle n|\hat{\rho}(\mathbf{q}-\mathbf{G},z_2)|m\rangle e^{-|\mathbf{q}-\mathbf{G}||z_1+z_2|} P_m \right] \delta(E - E_n + E_m). \tag{20}$$

where

$$\sigma_0 = \frac{V^2 m^2 |R_{\mathbf{G}}|^2 (N_s N_i)^2}{2\pi\hbar^4} \sqrt{\frac{E_f}{E_i}}. \tag{21}$$

This result is what is used in Section 4 of the main manuscript.

The main conclusion here is that the Coulomb matrix element in Eq. 20, and hence the M-EELS cross section, is enhanced whenever the momentum transfer coincides with a structural periodicity in the material, i.e., whenever a Bragg condition is met. This conclusion was not reached by earlier authors, who considered only the case of a surface that is translationally invariant [8, 12, 13, 44]. If the experiment is carried out in near-specular conditions, i.e., with $\mathbf{q} \sim 0$, one can drop all but the $\mathbf{G} = 0$ term in the sum. In this case, assuming further that the reflection coefficient $R_0$ is real, one recovers the Mills result,

$$\frac{\partial^2 \sigma}{\partial \Omega \partial E} = \sigma_0 \sum_{m,n} V_{\text{eff}}^2(\mathbf{q}) \int_{-\infty}^0 dz_1 dz_2$$
$$\times \left[ \langle m|\hat{\rho}^*(\mathbf{q},z_1)|n\rangle \langle n|\hat{\rho}(\mathbf{q},z_2)|m\rangle e^{-|\mathbf{q}||z_1+z_2|} P_m \right] \delta(E - E_n + E_m). \tag{22}$$

where

$$V_{\text{eff}}(\mathbf{q}) = \frac{4\pi e^2}{q^2 + (k_i^z + k_s^z)^2}. \tag{23}$$

This expression applies to the case of what was traditionally called "dipole scattering," in which the semiclassical trajectory of the probe electron keeps a significant distance from the surface [8].

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
