# Peer review of "Measurement of the dynamic charge response of materials using low-energy, momentum-resolved electron energy-loss spectroscopy (M-EELS)"

_SciPost Physics, doi:SciPost Phys. 3, 026 (2017)_

## Round 4 · Referee Report · Anonymous · 2017-8-15

Strengths

1) Compared to other recent papers on momentum resolved EELS, this work contains a far more substantial introduction and theoretical background relating the experimental data to the dynamic charge susceptibility.

2) The paper is exceptionally well written.

3) The authors deserve credit for relating EELS data to more widely used momentum resolved spectroscopies, in particular ARPES.

4) The paper is likely to stimulate progress in correlated electron physics.

Weaknesses

1) The paper does not provide significant new insight into the material under investigation.

2) The paper oversells the experimental advances made by this group. From reading abstract and introduction alone, one gets the impression that the authors would have invented momentum resolved EELS. However, this is completely wrong. Momentum resolved EELS with spectrometers of the same type is used in the surface science community since 2 decades (see e.g. PRB 61, 16911 (2000) as an example). Moreover, other groups have recently implemented technically more advanced approaches to momentum resolved EELS (Refs. 38, 39). Later sections of the paper acknowledge this work more or less appropriately but from the way the first two pages are written now, many readers will get a completely wrong impression of the history and current status of EELS from this paper.

Report

Vig et al. apply low-energy EELS to derive the dynamic charge response function of optimally doped Bi2212. This topic is of significant interest for many scientists working on "quantum materials". The experimental data is of good quality and its analysis is scientifically sound. On the other hand, the main results of the paper, namely the identification of different phonon modes and the observation of a plasmon near 1 eV, are not new and do not significantly advance our understanding of high-Tc cuprates.

In my opinion, the key contribution of this paper is the theoretical part relating EELS data to quantities of interest for correlated electron physics. This offers plenty of food for thoughts and makes the work interesting for a large community. At the same time, it provides arguably the most comprehensive guide to the interpretation of EELS data to date and will be an extremely valuable resource for newcomers trying to get into this field.

Requested changes

1) I suggest to rephrase the sentence "Here, we demonstrate a may to measure chi by applying momentum-resolution methods from x-ray and neutron scattering to surface HR-EELS" in the abstract and a similar sentence in the introduction to avoid giving the impression that the present work introduces a new technique.

---

## Round 4 · Referee Report · Anonymous · 2017-8-25

Strengths

1- the paper is very well written provide a detailed and highly valuable theoretical background, much better than those I came across in the existing in the litterature.
2- Whereas the technique has been around for some time, it has only marginally been applied to correlated systems. The present works highlights this very well and a strong case for the method is made.

Weaknesses

1- Some inaccuracies. I am surprised by the statement 'The problem with EELS is that meV energy resolution has not yet been demonstrated in an instrument that is also momentum-resolved.' Such devices have been developped in Germany a few years ago (see e.g. http://journals.aps.org/prl/abstract/10.1103/PhysRevLett.104.137203 or this still recent review http://dx.doi.org/10.1016/j.physrep.2014.08.001, where high-resolution momentum resolved EELS has been used to study magnetic excitations in ultrathin films. Arguably these experimental set-ups were not developed to measure charge dynamics (in fact spin polarization seem to be an extra complication), I believe they represent concrete examples of momentum-resolved EELS with meV energy resolution. I am surprise not to see such examples mentioned in the current paper.

2- To the best of my knowledge the statement 'Despite steady improvements, however, RIXS techniques are still
limited to an energy resolution of ∆E ∼ 120 meV' is only valid for Cu L-edge RIXS. Edges of lighter atoms have higher resolutions.
Improved energy resolution has recently also been demonstrated (http://www.nature.com/nphys/journal/vaop/ncurrent/full/nphys4157.html). I agree however with the authors that the cross-section issue is a strong caveat of this method.

3- EELS is an highly surface sensitive technique. Here the authors use relatively small energies and work in transmission. I´d expect they have to work on extremely thin layers of material, yet I haven´t found anything regarding the actual sample thickness.

4- Not much is learnt about the studied material

Report

Overall I believe this is a very useful paper that should motivate subsequent work.
Demonstration of a table-top high energy resolution and momentum resolved probe of the charge dynamics, and this opens new exciting perspectives. The theoretical background and the method are so well described that, despite maybe limited novelty regarding the physics of the cuprates, I am convinced that this paper will serve as a reference for further HR-EELS studies for correlated materials.

Requested changes

see the remarks in the weaknesses section above.

---

## Round 5 · Author Response

GENERAL REMARKS

We are resubmitting our paper “Measurement of the dynamic charge response of materials using low-energy, momentum-resolved electron energy-loss spectroscopy (M-EELS)” for publication in SciPost. Both reviewers praised the value of our demonstration of a q-resolved probe of the dynamic charge response of materials, as well as the quality and clarity of the manuscript itself, which both argue will become a reference for the field.

However, the reviewers raised two criticisms. The first was that we did not learn anything new about cuprates from this study. We do not deny this. The purpose of this manuscript was to explain how to do surface EELS experiments with true momentum resolution, to explain how the scattering cross section is related to the dynamic charge response, and to validate these claims with measurements. The new physics here is in the understanding of the technique itself, BSCCO just serving as a case study. For this reason, we believe the manuscript is highly deserving of publication, despite the absence of new insight into the material itself.

The second criticism is more fundamental and needs to be addressed. The central claim of our paper is that, after four years of development, we succeeded in making the HR-EELS technique fully momentum-resolved so that it can do true, reciprocal space measurements. We call this new method “M-EELS”. Both reviewers, particularly #1, argued that we are overclaiming because HR-EELS was already a momentum-resolved probe. The implication is that our M-EELS setup is the same as an HR-EELS setup, and we are just doing things in the conventional way. This represents a very serious misunderstanding of our experiment and one of the most important points of our paper, so it is crucial we clarify this.

It is useful to start by reflecting on what it means for an experimental technique to be “momentum-resolved.” We propose the following definition: A technique is momentum-resolved if it measures some spectroscopic quantity, such as a correlation function or spectral function, at a defined location in momentum space. Such a technique must have two attributes:

  1. It must probe a small momentum volume (i.e., it must not be q-integrating)
  2. The location of this volume in momentum space, q, must be controllable and known.

Consider, for example, the case of ultraviolet photoemission spectroscopy (UPS), which is a commercially available surface science technique that is not considered momentum-resolved. UPS is used for chemical analysis of surfaces, e.g., determining elemental composition, binding energies, core level shifts, etc. These are not momentum-dependent quantities.

UPS is not a momentum-integrating probe, however. A commercial UPS analyzer samples a relatively small range of k space. Nevertheless, this technique is not considered momentum-resolved because UPS setups are designed to work at fixed angle. It is common practice to get some momentum information in UPS by rotating the sample on a stick, which can reveal dispersing energy bands. But researchers do not consider this to be true momentum scanning because k is not known precisely. In other words, UPS satisfies attribute #1 above, but not attribute #2.

Instruments exist, however, that can quantitatively measure the momenta of photoelectrons. This is accomplished by using an angle-resolved electron analyzer in conjunction with a high-resolution, multi-axis goniometer for the sample, a technique called angle-resolved photoemission spectroscopy (ARPES). ARPES is based on exactly the same photoelectric effect as UPS, but is considered a different technique because it can measure momenta with an accuracy better than 1% of a Brillouin zone. This additional capability enables APRES to measure Fermi surfaces, for example, which is not possible with simple UPS.

Returning the discussion to scattering, HR-EELS instruments work in a manner similar to UPS. In the case of an Ibach setup, for example, the analyzer is set at a fixed scattering angle (using a hand crank) and the emphasis is on measuring the meV excitations of surfaces that have been coated with interesting thin films, dosed with molecular adsorbates, etc.. Such measurements yield extremely rich information about surface properties.

Like UPS, HR-EELS is not momentum-integrating, and it is common practice to get some momentum information by rotating the sample on a stick. One can see dispersing phonons and plasmons in this manner, just as one can see dispersing bands in UPS. Nevertheless, HR-EELS cannot be considered a q-resolved probe for the same reason UPS is not: the momenta are not measured to a sufficiently high degree of accuracy. For example, just as UPS cannot measure Fermi surfaces, HR-EELS cannot measure diffraction patterns. Like UPS, HR-EELS satisfies attribute #1 above, but not attribute #2.

The purpose of M-EELS is to do for HR-EELS what ARPES did for UPS: raise the accuracy with which momenta are measured so it can function as a true, momentum-resolved probe. As explained in Section 3 of our paper, mounting a hemispherical analyzer on the chamber does not guarantee success. A fundamental difference between photoemission and EELS is that, in the latter, the incident beam carries significant momentum (the photons in ARPES have k~0), so the electron trajectories must be referenced not only to the crystal axes, but also to the direct beam. Small errors in these angles can lead to large errors in the momentum, q.

For this reason, M-EELS takes its inspiration not from ARPES, but from x-ray and neutron scattering, which are designed to reference the angles of the scattered particles to the incident beam. As described in Section 3 in our paper, this involves many additional degrees of freedom--mostly for centering the axes of rotation--that are not present in conventional, HR-EELS setups. Mimicking the degrees of freedom of a triple axis neutron spectrometer, our M-EELS setup can determine the momentum transfer of the scattered electrons with an absolute accuracy of ~0.01 angstrom-1, which for BSCCO is better than 1% of a Brillouin zone. Our M-EELS also employs a new, novel type of control system, adapted from synchrotron beamlines, which enables coordinated motions of the lens voltages, sample goniometer angles, and the analyzer scattering angle, to keep the momentum fixed during energy scans.

Just as ARPES can measure Fermi surfaces that UPS cannot, M-EELS can measure diffraction patterns that HR-EELS cannot. This is illustrated in Fig. 4 of the current manuscript, which shows a zero-loss map of reciprocal space showing the (0,0) and (1,0) Bragg peaks of BSCCO, as well as the supermodulation reflections along the (1,1) direction. Another example is a map of the order parameter reflections in the CDW material, TiSe2, shown in Fig. 2a of arXiv:1611.04217. Such maps are a unique capability of M-EELS and illustrate the special capabilities of a technique that is truly q-resolved.

RESPONSES TO SPECIFIC REFEREE CRITICISMS

RESPONSE TO REFEREE #2

1- Some inaccuracies. I am surprised by the statement 'The problem with EELS is that meV energy resolution has not yet been demonstrated in an instrument that is also momentum-resolved.' Such devices have been developped in Germany a few years ago (see e.g. http://journals.aps.org/prl/abstract/10.1103/PhysRevLett.104.137203 or this still recent review http://dx.doi.org/10.1016/j.physrep.2014.08.001, where high-resolution momentum resolved EELS has been used to study magnetic excitations in ultrathin films. Arguably these experimental set-ups were not developed to measure charge dynamics (in fact spin polarization seem to be an extra complication), I believe they represent concrete examples of momentum-resolved EELS with meV energy resolution. I am surprise not to see such examples mentioned in the current paper.

OUR REPLY: The devices the referee cites are Ibach-type HR-EELS spectrometers, identical to the one we used in our study. The only difference is that the above studies employ a spin-polarized source to allow them to study magnetic excitations, for which the authors use the acronym “SP-EELS.” These instruments work at a fixed angle and are not q-resolved in the sense defined above. Our manuscript describes a different variant on HR-EELS that is momentum-resolved, for which we propose the acronym M-EELS. The above references are superseded by Refs. 7 and 8 in our current manuscript.

CHANGES MADE: None.

2- To the best of my knowledge the statement 'Despite steady improvements, however, RIXS techniques are still limited to an energy resolution of ∆E ∼ 120 meV' is only valid for Cu L-edge RIXS. Edges of lighter atoms have higher resolutions. Improved energy resolution has recently also been demonstrated (http://www.nature.com/nphys/journal/vaop/ncurrent/full/nphys4157.html). I agree however with the authors that the cross-section issue is a strong caveat of this method.

OUR REPLY: The referee has raised a good point. The resolution achieved with RIXS has improved even since the first version of our manuscript.

CHANGES MADE: We removed our earlier RIXS reference and replaced it with one demonstrating 40 meV, making appropriate adjustments to the text. Note that this resolution is still 20x worse than what we can achieve with our M-EELS setup.

3- EELS is an highly surface sensitive technique. Here the authors use relatively small energies and work in transmission. I´d expect they have to work on extremely thin layers of material, yet I haven´t found anything regarding the actual sample thickness.

OUR REPLY: Our experiment actually works in reflection geometry. The sample thickness is therefore not relevant, though it is typically 0.1 mm.

CHANGES MADE: None.

4- Not much is learnt about the studied material

OUR REPLY: We agree with this criticism. The new physics is our paper concerns the scattering technique itself. The BSCCO material was only a case study used to demonstrate the capabilities of the technique. While not much new was learned about the material, we believe the paper merits publication based on the scattering physics alone.

CHANGES MADE: None.

RESPONSE TO REFEREE #1

1) The paper does not provide significant new insight into the material under investigation.

OUR REPLY: This is true. The new physics here is in the scattering technique itself. We believe that this attribute alone merits publication. See the discussions above.

2) The paper oversells the experimental advances made by this group. From reading abstract and introduction alone, one gets the impression that the authors would have invented momentum resolved EELS. However, this is completely wrong. Momentum resolved EELS with spectrometers of the same type is used in the surface science community since 2 decades (see e.g. PRB 61, 16911 (2000) as an example). Moreover, other groups have recently implemented technically more advanced approaches to momentum resolved EELS (Refs. 38, 39). Later sections of the paper acknowledge this work more or less appropriately but from the way the first two pages are written now, many readers will get a completely wrong impression of the history and current status of EELS from this paper.

OUR REPLY: As explained above, HR-EELS is not a truly momentum-resolved technique. M-EELS is an enhancement to HR-EELS that allows the momentum of scattered electrons to be determined quantitatively to an accuracy of <1% of a Brillouin zone. M-EELS does for HR-EELS what ARPES did for UPS. Momentum maps, such as that shown in Fig. 4 of our manuscript, are not possible with HR-EELS, and can only be done with M-EELS. Note that neither Ref. 38 nor 39, which used hemispherical analyzers, was able to measure a diffraction pattern, which we routinely do with our setup. So the reviewer should be careful about claims of which approach is more “advanced.” Still, the referee has made a valid point that we did not invent the HR-EELS technique itself. We simply upgraded it to do quantitative momentum-space measurements. We agree that the tone of the introduction should be changed to more accurately reflect this.

CHANGES MADE: We modified the wording of the abstract, introduction, and Sections 2-3 to emphasize that our key advance was to upgrade the momentum accuracy of HR-EELS to bring it on par with other momentum-resolved probes, e.g., inelastic x-ray scattering, inelastic neutron scattering, and ARPES. We took great care to avoid leaving the impression of claiming more than this. We also added a paragraph that gives explicit numbers on our momentum accuracy and resolution, which clearly sets M-EELS apart from other techniques.

---

## Round 5 · List of Changes

We found it clearer to summarize the changes made in the reply to referee comments (see above).

You are currently on this page

Resubmission 1509.04230v5 on 15 September 2017

---

## Editorial Decision

published